# The Effect of a Parent-Directed Program to Improve Infants’ Motor Skills

**DOI:** 10.3390/ijerph20031999

**Published:** 2023-01-21

**Authors:** Marlene Rosager Lund Pedersen, Bjarne Ibsen, Danae Dinkel, Niels Christian Møller, Lise Hestbæk

**Affiliations:** 1Department of Sports Science and Clinical Biomechanics, University of Southern Denmark, Campusvej 55, 5230 Odense, Denmark; 2School of Health and Kinesiology, College of Education, Health, and Human Sciences, University of Nebraska at Omaha, H & K Building, 6001 Dodge Street, Omaha, NE 68182, USA

**Keywords:** motor skills, motor development, infants, parents, caregivers, parent-directed program, program, intervention, evaluation

## Abstract

Poor motor skills are associated with several factors that might delay children’s development. Therefore, early programs to promote a child’s motor development are essential. Within the first year of life, parents have a critical role in promoting their infant’s motor development. However, little research has explored parent-directed programs that promote infant development in a Scandinavian context. This study aimed to evaluate the effectiveness of a parent-directed program to improve infant motor development. Methods: Parents of infants received a parent-directed program that included guidance from health visitors on ways to promote motor development, videos with motor development activities and a bag with related materials. Two municipalities in Denmark took part in the study (one intervention, one control). Health visitors in both municipalities measured the infants’ age-appropriate motor skills once when the infants were between 9–11 months of age. A logistic regression model was used to analyze the data. Results: No difference was detected in motor development over time in the two municipalities regarding the proportion of children with age-appropriate motor skills. Conclusions: A parent-directed program in which parents were guided to play and encourage motor development with their infant showed no effect on infants’ age-appropriate motor skills at 9–11 months.

## 1. Introduction

Research suggests that young children with good motor skills are more likely to be physically active, participate in sports more often, and have better academic performance later in life than children showing less developed motor skills [1,2,3,4,5]. Furthermore, motor problems in a child’s early years (0–2 years) have led to an increased risk of motor problems when they start school [6,7]. Relatedly, improved motor skills in childhood have been linked to increased levels of leisure-time physical activity in young adults [8], which is associated with various mental and physical benefits [9]. Thus, there are good arguments for promoting children’s motor development, as it has many short-term and long-term effects.

Unfortunately, many children may not develop motor skills adequately [10]. For example, a study in Denmark of 16,686 infants born in 2017 and aged 8–10 months found that 10% received a "comment" from their health visitor regarding their motor development, indicating a potential delay [10]. However, the proportion who received a "comment" varied from municipality to municipality. Efforts are needed to ensure optimal achievement of motor skills for all infants.

Parents play an essential role in improving infant motor development by being role models; providing opportunities for movement and support [11,12,13,14,15]; and preventing the infants from being sedentary, such as by placing them in a restrictive device [15,16]. Improving an infant’s motor development requires adult involvement, as they must implement daily activities to promote motor development. For example, the parents may place an infant on his/her stomach [15,17,18], as this strengthens the infant’s muscles for motor milestones such as controlling the head, reaching, crawling, and pulling him/herself up [15]. However, previous research suggests that parents may be unaware of recommendations for encouraging movement and motor development [19,20]. Relatedly, studies show that parents of young children tend to overestimate their young child’s motor competence and development by rating their motor skills better than they are [21,22]. This could be because parents are not familiar with what the child should be able to do [21]. Importantly, parents desire more specific guidance and information on promoting infant movement and motor development [23]. However, other studies showed that parents provided dependable accounts of their infant’s motor development [24,25,26]. In addition to being measured by parents, an infant’s motor skills can also be reported by a trained examiner by, for example, developmental assessments such as the Bayley Scales of Infant Development [27]. However, in Denmark, many municipalities use an overall measure of whether the infant’s motor development is age-appropriate based on an assessment of nine motor milestones, which the health visitors score in the infant’s familiar surroundings at home with the parents [28]. This assessment, in Denmark, is an overall measure of the infant’s motor development at age 9–11 months that tells whether the child has age-appropriate motor skills or not. This test is more limited in the score compared to detailed assessments such as the parent-reported questionnaire EMQ [24] or Alberta Infants score [29].

Previous studies have shown parent-directed interventions to be effective [23,30,31] and that parents can gain skills that help to strengthen their child’s motor development and understand its importance [15,32]. These studies have not been investigated in a Scandinavian context. Such knowledge and skills to stimulate infants’ motor development can be obtained in Denmark through the Danish home visiting program. This program has been implemented nationally since 1974 and consists of having a health visitor visit parents five times at minimum over the child’s first year of life [33,34]. However, the final number of visits is individual and varies from municipality to municipality. The health visitor’s goal is to guide and advise parents on their child’s development, including motor skills.

A recent review of interventions focused on infant motor development found several elements essential to these interventions [23]. These components included: (1) supervision and guidance for the infant’s caregivers to acquire knowledge and skills about age-appropriate motor skills; (2) age-appropriate training to build up parents’ skills for promoting motor development; (3) inspiration for activities and ways to play with their infant to stimulate the infant’s motor development; and (4) knowledge about age-appropriate toys [23]. From those previous results, we wanted to test some of these elements in a large community-based study.

Therefore, a Danish municipality decided to introduce a new program to improve infants’ motor skills through a "parent-directed program" designed to develop the competencies and skills of parents to promote their infant’s motor skills. The health visitors implemented the program during their regular home visits. The regular home visits and the parent-directed program will be explained in further detail in the Methods section.

This study aimed to evaluate the parent-directed program’s effect on infants’ age-appropriate motor development at the age of 9–11 months. Our hypothesis was that the parent-directed program would affect infants’ motor skills in a positive way compared to those of infants in a comparison municipality.

## 2. Methods

### 2.1. The Context: The Home Visits and the Health Visitors

In Denmark, health visitors’ efforts to promote infants’ health and well-being are central to the government’s programs to ensure infants have the best start in life [35]. A key element is the home visiting program, in which health visitors supervise and guide parents on infants’ development, including motor development. In Denmark, health visitors are a specialized professional group authorized to handle tasks related to child health and are trained on how infants and children should develop motor skills [22]. To become a health visitor, you must be an educated nurse, after which you must have a minimum of two years of experience working with children. Thereafter, you receive 18 months of theoretical and practical training. Health visitors see many infants and, therefore, have extensive experience identifying infants who do not develop age-appropriate motor skills [36].

The Danish home visiting program is very well accepted by parents [37], and only one or two families out of 1000 reject contact with a health visitor [34]. Denmark’s municipalities offer home visits, but each municipality decides on the final number of home visits [34]. The selected municipalities in our study had six mandatory home visits, starting when the child was 4–5 days old, with follow-up visits at the ages of 7–10 days, 3 weeks, 2–3 months, 4–6 months, and 9–11 months. All families are assigned a health visitor who follows the child throughout the program to provide continuity.

### 2.2. Study Design

This study investigated infants’ motor skills before and after a parent-directed program in a municipality in Denmark to improve infants’ motor skills. This intervention municipality was selected based on the planned implementation of the parent-directed program. A second municipality, which did not use the program, was used as a comparison group. All participants were infants 9–11 months old within the intervention or comparison municipality. Informed consent was obtained prior to participation in the study. The study was approved by the Institutional Review Board of the University of Southern Denmark, Research and Innovation Organization (R.I.O.) (protocol code: 11.176). All reports of data were anonymized.

### 2.3. Study Populations

The intervention was implemented in Hoeje-Taastrup Municipality. All parents of infants born in Hoeje-Taastrup Municipality from January 2020 to October 2022 received the intervention described in more detail below. Infants who were 9–11 months old in 2021–2022 (and the last three months of 2020) and whose parents participated in the entire intervention were included in the study. Therefore, the cohort comprised infants aged 9–11 months who had undergone the intervention from October 2020 to October 2022 (*n =* 1282). Throughout the paper, this period will be mentioned as 2021–2022, knowing that October, November, and December of 2020 were included. In order to be able to compare the motor development of children within the intervention municipality over time, data from prior to the intervention were also extracted. Thus, data from all infants aged 9–11 months from January 2019 to September 2020 in the Hoeje-Taastrup Municipality (*n =* 403) were included in the data analysis.

The comparison municipality was selected based on sociodemographic conditions that were relatively similar to those of the intervention municipality, obtained from Statistics Denmark [38,39] below: (1) education—the highest completed education (15–69 years) by time, highest completed education and area of residence; (2) income—disposable income by unit, time, income range and area; (3) ethnicity—population in the 1st quarter of 2020 by time, origin and area; (4) socio-economic status—the highest completed education (15–69 years) by time, socio-economic status and area of residence; (5) demographic similarities; and (6) a similar structure for home visits. The most significant difference between the two municipalities was that the intervention municipality had twice as many inhabitants as the comparison municipality. All infants aged 9–11 months from October 2020 to October 2022 were included as participants from the comparison municipality (*n =* 509). Additionally, similarly to the intervention municipality, data on the infants’ motor skills before the intervention were extracted to compare the motor development of children in the comparison municipality over time. All infants aged 9–11 months from January 2019 to September 2020 in the comparison municipality (*n =* 242) were also included for data analysis.

A description of the number of infants from the two municipalities included in the study at the two timepoints can be seen in Table 1.

### 2.4. The Intervention Program

From January 2020 until October 2022, all parents in Hoeje-Taastrup Municipality (called intervention municipality throughout the article) underwent a parent-directed program designed to develop the competencies and skills to promote their infants’ motor skills. The health visitors implemented the program during regular home visits (*n =* 6), which occur when the child is aged 4–5 days, 7–10 days, 3 weeks, 2–3 months, 4–6 months and 9–11 months [40]. The parent-directed program consisted of three elements. First, health visitors received a practical and theoretical course on motor development to strengthen their competencies. With this knowledge and insight into infants’ motor development, the health visitors were instructed to guide parents by providing suggestions for activities and ways to play with their infants as well as information about factors inhibiting motor development. The scope and nature of the guidance varied from infant to infant, determined by the need for guidance. Second, all new parents received a bag with motor-stimulating toys (see Figure 1) from the health visitors when their child was 2–3 months of age with instructions on how they could be used for play and activities. Third, the health visitors handed out videos with motor activities and ideas for games for parents to play with their infants and to inspire parents to practice these activities with their infants. The time each health visitor spent teaching and guiding the parents about motor development was not formalized and could differ between visits. These three parts and their implementation are further described in Table 2.

No extra intervention regarding motor development and activities that promoted motor development were added to the regular home visits by health visitors in the comparison municipality. The parents of infants received only the usual guidance with the same six mandatory home visits, which occurred when the child was at the same age as those visited in the intervention municipality. The usual guidance consisted of information about the time after birth, breastfeeding or nutrition for the infant, the child’s development, and the parents’ role as both first-time and multiple-time parents [33]. During the visits, the health visitors assessed the child’s health, well-being and development.

### 2.5. Variables

Outcome: The health visitors observed all infants’ motor development during the mandatory home visits when infants were 9–11 months [28]. The test was based on these motor development milestones [10]: its by him/herself; rolls from back to stomach; supports flat feet; creeping—about to crawl; begins to stand up with support; better control of hands and fingers, holding on to objects and letting go; develops pincer grip; chews food with a coarser consistency and takes an interest in eating by him/herself; waving and clapping towards the end of the period [10]. In Denmark, this is a widespread national test used by 72 out of 98 municipalities to assess an infant’s motor development [28,41], where the health visitor scores the infant’s motor development at the age of 9–11 months [28]. The health visitors are, therefore, accustomed to using the test in practice. The test was not developed to measure finer nuances and differences in motor development but to assess infants at great risk for motor development delay [28,41].

According to these nine motor development milestones, the health visitors assessed whether the infant’s development was age-appropriate (A), required attention (O), or whether extra intervention was needed (I) [28,41]. The health visitors were unaware of the use of this study, as they always observed and scored the infant’s motor development.

This scoring of motor development was stored in a national database [42] and used as the outcome of this study. Motor development was dichotomized into age-appropriate (A) or not age-appropriate (O+I).

Covariates: Age was calculated in days at the time of each test for age-appropriate motor skills. The gestational age was calculated as the week the infant was born. Year was either 2019–2020 before or 2021–2022 after the intervention. Municipality was either the comparison municipality or the intervention municipality.

### 2.6. Statistical Analysis

We presented crude descriptive data for both municipalities across year 2019–20 and 2021–22, respectively. Descriptive data were presented by group as means and standard deviations (SD) for each group for continuous variables. The motor development scores were presented as proportions of children with age-appropriate development.

First, after stratifying for municipality, we used a logistic regression model to investigate the age-appropriate motor skills scores within each municipality. The dependent variable was age-appropriate motor skills (0 = no; 1 = yes) and the independent variable was year of assessment (0 = 2019–20; 1 = 2021–22) A priori selected covariates were adjusted for in the model in the form of gestational week, and children’s age at test in days.

We then included the municipality variable (0 = comparison municipality; 1 = intervention municipality) as another independent variable in the model and also added an interaction term between the year of assessment and municipality to examine if children’s motor skills developed differently over time in the intervention group compared to the control group.

We tested for inflated regression coefficients by use of the Collin’s test in Stata, where values larger than 10 indicate a high level of multicollinearity [43]. All variation inflation factors were found to be below 3.35 observed in our models, indicating no violation of model assumptions. We used the goodness-of-fit test post doc to test how well the model fit [44]. The model fit without evidence of model deficiency (*p* = 0.23).

Post hoc, we tested if the intervention program affected preterm infants in particular. We used the same model described above but restricted the analysis to infants born preterm according to the WHO definition of preterm infants. The definition of preterm birth is <37 weeks [45]. All data were analyzed with Stata (version 17.0) software, and statistical significance was set at *p* < 0.05.

## 3. Results

The descriptive data are shown in Table 3. In total, 2336 infants aged 9–11 months were included from the two municipalities at the two timepoints, with most infants being included in the intervention group (*n =* 1182). There were no significant differences between the pre-intervention and intervention periods regarding children’s age-appropriate motor skills and age when tested. More children in the intervention municipality had age-appropriate motor skills compared to children in the comparison municipality—both before the start and at the end of the intervention. The comparison municipality displayed a larger increase than the intervention municipality, from the pre-intervention period to after the intervention, in the proportion of children with age-appropriate motor development. However, the increase was not significant for either municipality (*p* = 0.95 and *p* = 0.46, respectively).

The results of the logistic regression model applied after stratifying on municipality to investigate the age-appropriate motor skills scores before and after the intervention period within each municipality are shown below in Figure 2.

The odds of having age-appropriate motor skills were higher (OR = 1.25) in 2021–2022 compared to 2019–2020 among children in the comparison municipality; however, this change was not statistically significant (*p* = 0.41). No changes occurred over time among children in the intervention municipality regarding the odds of possessing age-appropriate motor skills (OR = 0.99 and *p* = 0.97).

The adjusted logistic regression model with the interaction term is shown in Table 4. The results show an odds ratio of OR = 0.8, *p* = 0.53 for interaction between intervention and municipality, meaning that children’s odds of having age-appropriate motor skills did not change differently over time across the two municipalities, indicating no effect of implementing the intervention program.

The odds of having age-appropriate motor skills increased by 27% every week as the gestation period was extended (*p* < 0.00) for both municipalities. Going from the comparison municipality to the intervention municipality, the odds of having age-appropriate motor skills were OR = 2.13 (*p* = 0.012) for both years.

The results showed no effect of the intervention on children’s age-appropriate motor skills when the analysis was restricted only to include infants born preterm. The results showed an odds ratio of OR = 1.26, *p* = 0.82 for interaction between year of assessment and municipality, meaning no difference in the change of proportion of preterm infants with age-appropriate motor skills was observed across the two municipalities over time.

## 4. Discussion

### 4.1. Summary of Findings

To our knowledge, this is the first study in Scandinavia investigating the effect of a parent-directed program on the motor development of a large sample of infants aged 9–11 months. Our study showed no acute effect after the intervention on the infants’ age-appropriate motor skills. Furthermore, no effect was found when looking at preterm infants. We cannot conclude that the parent-directed program influences an infant’s age-appropriate motor skills when the infant is 9–11 months old.

### 4.2. Comparison with the Literature

Little research has been published on community-wide interventions promoting infants’ motor development, compared to children of, for example, kindergarten age and preschoolers [46]. Previous research in infants has primarily focused on those born preterm [47,48]. While changes in an infant’s motor development can be difficult to investigate, as infants naturally have a relatively progressive development in the first year of their life, it does not change the fact that it is important to identify programs that can help promote infant motor development.

Several studies have been conducted on infants of the same age, but with small samples and not in a Scandinavian welfare context. Our findings do not align with earlier research with programs focused on improving parents’ knowledge and skills to increase infants’ motor development [31,49,50]. For example, studies investigating interventions on infants under one year, where the caregivers must play and perform activities with their infants, show that the interventions affect the children’s motor development [31,49,50]. However, the samples in two of these studies were small (*n =* 28 and *n =* 7), and they measured whether infants were meeting specific motor milestones (head control and tummy time). However, they used no dichotomous variable and were able to look at finer differences. Our study was different, as we had an overall measure of whether the infant’s motor skills were age-appropriate or not. However, these three interventions used many of the same elements: stimulating toys (grasp ball and rattle), activity and play ideas, and supervision [31,49,50]. Our study was designed in a way that was realistic to implement on a large scale, in contrast to other, more intensive interventions that, for both practical and economic reasons, would be challenging to implement on a large scale.

Another study included a more extended intervention period from birth until the child was 33 months old [51]. This study showed that the intervention group was less likely to have motor development delays than the control group. In this study [51], the parents received guidance on activities, teaching and play to promote the infant’s motor development. They also received developmental materials, e.g., toys [51]. This may suggest that a longer intervention may be needed to see improvements in the current study.

### 4.3. Methodological Considerations Concerning Our Study

The methodological strengths of our study were the relatively large sample in the intervention group and ability to compare the scores with another municipality. Furthermore, our sample was not a selected group but all infants in the municipalities where the health visitor had remembered to register the child’s motor skills in the record system; only one or two families out of 1000 rejected contact with the health visitor [34]. Secondly, a strength was that the infants were observed and tested in their usual and familiar surroundings at home. Thirdly, it was a strength that experienced health visitors observed and scored the infants using a measure that is widely used within a real-world setting. Fourthly, it was also a strength that the health visitors, on the basis of their great professionalism and experience, gave an overall assessment of each infant’s motor development. A fifth strength was that the health visitors were unaware that we used their scores on infants’ motor development; thus, measurement bias was not an issue.

However, the study also has methodological weaknesses. Firstly, it can be criticized for not using a test where a study has tested the reliability and validity of the test. Our previous review [5] showed that there is not only one way to measure whether a young child has improved motor development, and in the literature, different methods were used for this. We used a widespread test in Denmark, which 72 out of 98 municipalities used for children’s developmental assessments. However, the test has not been published and validated so far. On the other hand, the method enables the analysis to include a large group of children, unlike other methods, which would be challenging to use on so many children given the constraints on health visitors’ time. Further, it is a test that assesses whether there is a large risk for motor development delay. The test also allowed the comparison of results with previous years, as it had been used for the years before the intervention period this data was used for justification for the need for this study. However, future studies should explore other measures that could be widely utilized to examine finer changes in the development of specific motor skills.

Secondly, another potential limitation was that our study was not an RCT where the groups were matched from the beginning. In our study, the intervention municipality was selected based on the planned implementation of the parent-directed program. The comparison municipality was chosen because it was the municipality that most resembled the intervention municipality and, therefore, was not randomized. However, given the setting, an RCT would not have been feasible, and we believe an appropriate comparison group was utilized.

Thirdly, we use a dichotomous outcome variable. The test does not reveal if there were any improvements within the normal range of age-appropriate motor skills. It is up to each health visitor to score the child, but precisely what the individual health visitor emphasizes, plus the potential differences between the health visitors, can influence the scores, and observer reliability testing was not conducted. However, the health visitors have extensive experience and expertise with infants and their motor development and if it deviates from the norm.

Fourthly, the different potential impacts of dose response can be criticized. In the parent-directed program, the parents do not have a manual indicating how much they should implement activities with their infant. The parents found the program feasible in terms of implementation, which sharpened their focus on stimulating their infants from an early age [52]. However, the dosage can be very different, and we were unable to measure how intensively the health visitors or parents implemented the suggested activities. It is a limitation that we did not know the actual duration in which parents engaged with the materials. A review also showed a consensus on how the dose and response should affect the infant’s motor development [23]. For example, the parents should perform the activities for ten minutes each day with different toys, or tummy-time of 30 min per day. This could be one of the explanations for why we did not see an effect of the intervention.

### 4.4. Other Methodological Considerations

Regarding our study, a relatively high percentage of infants in 2019–2020 already had age-appropriate motor skills. Thus, it was more challenging to show improvement primarily due to using a dichotomous variable. We could only access data from the data register when the parent-directed program ended, and we were unaware of the percentage of infants who did not have age-appropriate motor skills in these years. We were unaware since we did not have free access to the data register. We only knew that in 2016, 24.1 percent of infants aged 9–11 months in the intervention municipality did not have age-appropriate motor skills [10]. This estimate was relatively different from the approximately 10 percent who do not have age-appropriate motor development in 2019–2020 (Table 3).

### 4.5. Perspectives

From a research perspective, one could, in the future, use a standardized tool for evaluating motor development, such as the Alberta Infant Motor Scale (AIMS) [53], which is validated, reliable, and test-specific for motor skills [29,54,55]. A matched design with an RCT—with fewer infants—could implement the program and be tested with, for example, AIMS. If it shows an effect, a scale-up of the study could be relevant. Additionally, future research should measure the implementation of the suggested activities by parents, in order to learn how much time, the parents spend with the activities in everyday life with their children.

From a health perspective, it is still relevant to guide parents and provide them with knowledge and ideas regarding how to play with and stimulate their infants—especially as this has been a reported desire of parents in previous studies [19,20]. In future set-ups, selecting a group of infants who have already been identified as having a motor delay could be more beneficial, as these infants and their parents may especially be in need of motor development education. Furthermore, it could also be relevant to see if there is an effect later in the child’s life. This study identified no acute effect after the intervention, but perhaps these children may have better motor skills upon enrolling in school, so the given effect would be seen only later.

## 5. Conclusions

In conclusion, the program did not show an acute impact on the age-appropriate motor skills of infants in the intervention group, who had undergone a parent-directed program where parents received knowledge, stimulating toys, and supervision regarding activities they could perform with their infants. Due to the global measurement used in this study, future research could investigate if the intervention affected specific motor skills. It is possible that the intervention may have improved specific motor skills but that this was not detectable using a global rating.

## Figures and Tables

**Figure 1 ijerph-20-01999-f001:**
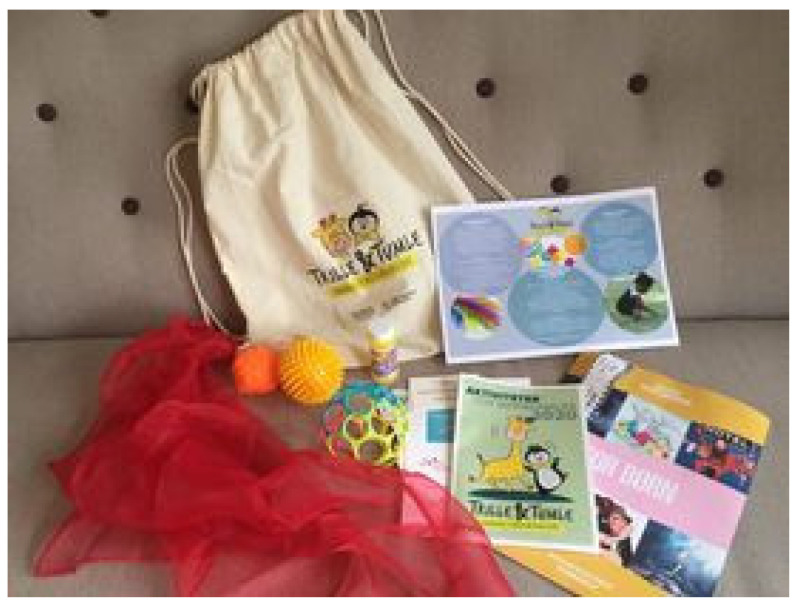
A bag with motor toys every parent received.

**Figure 2 ijerph-20-01999-f002:**
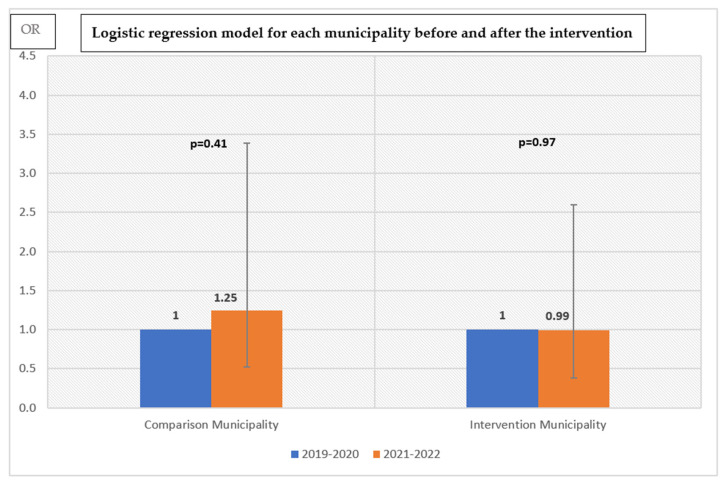
Odds ratios of children having age-appropriate motor skills. Results are from logistic regression models stratified in municipality and adjusted for the year of assessment, gestational week, and children’s age at test in days. OR with 95% confidence interval.

**Table 1 ijerph-20-01999-t001:** The different groups of infants divided into year and municipality are shown in this table. The first data collection in both municipalities was in 2019–2020, with no intervention. The second assessment was in 2021–2022 on all infants aged 9–11 months living in these two municipalities, where the infants in Hoeje-Taastrup Municipality had undergone the municipality-based parent-directed program.

		Year	
		2019–2020	2021–2022
Municipality	Hoeje-Taastrup Municipality	(*n* = 403)	Intervention group (*n* = 1.182)
	Comparison Municipality	(*n* = 242)	(*n* = 509)

**Table 2 ijerph-20-01999-t002:** Description of each element of the parent-directed program.

The Parent-Directed Program		
	Description of Each Element	Implementation with the Parents
Competence development of the health visitors	The health visitors took part in a competence development course on motor development. The motor development course consisted of six lessons, each lasting three hours. The first two times, the course content was about motor skills in everyday life; the third and fourth times, the content was about tumble play; and the last two times, the content was about presence and calm. An expert in the field taught the health visitors. The courses consisted of a combination of theory and practice. In addition, there was a lesson on how to transfer this knowledge into the home visits with the parents of infants.	The health visitors provided knowledge on motor development, suggestions for activities, and information about factors inhibiting motor development.
A bag with motor toys	Each family received a bag with motor-stimulating toys, including a soap bubble, grip ball, massage ball, motor ball, and sensory scarf. The bag also included a description of how to use the materials, including ideas for play and exercises to promote both gross and fine motor skills. Each bag’s description included how parents could use the materials for slightly smaller and slightly larger infants. Here are a few examples out of many to the parents: throw the scarf in the air and let the infant try to catch it. Let the child swing the scarf around. Let the crawling child catch the balls. Let the standing child throw the balls. Play hide-and-seek with the soap bubbles.In addition, it included tickets to the municipal swimming pool and a brochure on activities for infants in the municipality of Hoeje-Taastrup.See Figure 1.	The bag was handed out to the parents in mandatory visit 4 when the infants were 2 to 3 months old (see Table 1). Each health visitor decided how to explain and demonstrate the use of the materials during the visits.
Short videos	Four short videos (approx. three minutes per video) were produced for the parents.Each video targeted a specific age group and referred to strengthening the child’s motor skills through play and activities: infants of 1–3 months, 3–6 months, 6–9 months, and 9–12 months.	Each health visitor informed the parents about the videos and where to find them during the age-appropriate visit (through a link, QR code, or website).

**Table 3 ijerph-20-01999-t003:** Characteristics of the infants in the four groups in the years 2019–2020 and 2021–2022.

	Intervention Municipality		*p*-Value	Comparison Municipality		*p*-Value
Groups	2019–2020 (*n =* 403)	2021–2022 (Intervention group) (*n =* 1.182)		2019–2020 (*n =* 242)	2021–2022 (*n =* 509)	
Age at test (days)	285.0 (27.1)	286.0 days (26.1)	0.60	284.1 (25.5)	284.4 (24.9)	0.89
Gestational age (weeks)	38.9 (2.2)	39.1 (2.0)	0.17	39.2 (1.8)	39.0 (1.9)	0.42
Age-appropriate motor skills (%)	93.05	93.15	0.95	88.43	90.18	0.46
Change in motor development over time	0.10			1.75		

*N =* 2336, Age at test and birth week values are mean (SD), age-appropriate motor skills distributed in percentage. The first row with groups in column 1 was all infants from the intervention municipality for 2019–2020, with no extra guidance regarding motor development besides the regular home visits. The first row with groups in column 2 shows all infants from the intervention municipality for 2021–2022 who had received the extra guidance in the regular home visits. In the first row with groups in column 3 was the comparison municipality with no extra guidance regarding motor development treatment besides the regular home visits in the years 2019–2020. The first row with groups in column 4 was the comparison municipality with no extra guidance regarding motor development besides the regular home visits in the same years as the intervention period, 2021–2022.

**Table 4 ijerph-20-01999-t004:** Logistic regression model adjusted for age at test in days, municipality, year, the birth week, and the interaction term between time and municipality (*n =* 2067).

	Adjusted Model for Age-Appropriate Motor Skills
Odds Ratio	*p*-Values	[95% Conf. Interval]
Gestational week	1.27	0.00	[1.20–1.35]
Age at test in days	1.00	0.21	[1.00–1.01]
Municipality	2.13	0.01	[1.18–3.86]
Year	1.23	0.43	[0.73–2.06]
Interaction between intervention and municipality	0.80	0.53	[0.39–1.62]
Constant	0.00	0.00	[0.00–0.00]
Goodness-of-fit testHosmer and Lemeshow		0.23

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
