# Peer review of "The Effect of a Parent-Directed Program to Improve Infants’ Motor Skills"

_ijerph, 2023, doi:10.3390/ijerph20031999_

Round 1

Reviewer 1 Report

Dear authors,

Thank you for the opportunity to review the manuscript entitled “The Effect of a Parent-Directed Program to Improve Infants' Motor Skills”.

I have minor comments to improve the manuscript and they are presented here:

General comments

Please do not use paragraphs with single sentences. Always choose to build the paragraph with two or more sentences, it helps the readers and the text flow easier (ex. Introduction - line 75).

Introduction

I think the introduction is well-written and accomplishes many aspects that will be investigated. Despite that, I believe that the parents' perception of children's motor competence and development could be used to help to explain the subject a little bit more. The literature has some papers published that could be interesting to point out. Here I put some examples:

Silva, S. D., Flôres, F. S., Corrêa, S. L., Cordovil, R., & Copetti, F. (2017). Mother’s perception of children’s motor development in southern Brazil. Perceptual and motor skills, 124(1), 72-85.

Cordovil, R., & Barreiros, J. (2010). Adults' perception of children's height and reaching capability. Acta psychologica, 135(1), 24-29.

At the end of the introduction section, please provide the investigation hypothesis.

Methods

2.7. Variables

This is my main concern about your investigation: “The test was based on these motor development milestones [33]: Sits by him/herself; Rolls from back to stomach; Supports flat feet; Creeping - about to crawl; Begins to stand up with support; Better control of hands and fingers, holding on to objects and letting go; Develops pincer grip; Chews food with a coarser consistency and takes an interest in eating himself; Waving and clapping towards the end of the period [33].” Being this test part of a national program and not being widespread internationally, please provide more information on how this test works. Also, this is a valid instrument? Why choose this test?

2.9. Statistical analysis

During this section, there are some words with different sizes. Please review all text to ensure that little problem doesn’t occur in other sections.

Discussion

Why separate this section into different subsections? It will be interesting to read the discussion with all the information united as one section.

Reviewer 2 Report

The manuscript entitled “The Effect of a Parent-Directed Program to Improve Infants' Motor Skills” reports findings on an interesting study examining the effects of a parent-education intervention implemented in Denmark. Strengths of the reported research include the large sample size and the population-based approach where all children in a geographic region were compared. However, the method to assess outcomes represents a major limitation and missed opportunity. Motor development is highly nuanced and encompasses several skills that emerge at different times and involve different muscle groups. The global assessment and scoring reported here does not do this complexity justice. The authors do discuss this limitation, but I do wonder if the available data could be re-scored in a more detailed fashion. Further comments and suggestions are provided below. Note, my comments go through the manuscript in order.

1)      Please rephrase “children with less motor skills” to “children showing less developed motor skills”

2)      The authors cite a research study [10] that showed that children across different municipalities varied in the likelihood of receiving a ‘comment’ regarding their motor development. This is curious. Why would children in one municipality be more likely to show poor motor development? What is different between these municipalities?

3)      The mere placement of a child into a restrictive device – such as car seat, high chair, stroller, or carrier – should not be identified as “inappropriate behavior”

4)      Thank you for the explanation on the education levels of the health visitors.

5)      How was the second municipality for comparison purposes selected? Was it similar in size? Please provide more details on how these two municipalities compare in a table rather than in footnotes.

6)      How much time did the health visitors spend on teaching parents about motor development? Was this formalized at all or was it informal and different for each family?

7)      The provided bag of toys seems to exclusively focus on fine motor skills such as reaching, grasping, and maybe throwing. None of the objects seem to encourage gross motor skills such as crawling or walking. Sitting could be encouraged with the provided materials if instructions encourage parents to place the child in a sitting position. Can you please provide more details here on what skills were meant to be encouraged by these materials?

8)      The writing of the Methods section should be improved. The splitting of “intervention” vs. “control” group is not as evident as it could/should be and the separate mentioning of participants across the two groups is confusing. This section needs to be re-written for clarity.

9)      The population in the intervention group seems more than twice as large than the population in the control group. The text mentioned similar sized municipalities. Please correct in the text.

10)   Is the motor assessment used standardized in any way? Is it based on an established and commonly used measure such as the AIMS, PDMS, Bayley, …?

11)   The use of a 0/1 dependent variable to assess motor skill outcomes is highly questionable. In the best case, I would predict no differences due to this approach – which is exactly what you find. Why not use a more nuanced variable? Your bag of toys only encouraged select motor skills, why not focus on adequate development in these sub-domains of motor development?

12)   That your results show a significant effect of gestation is not surprising. However, the significant effect of municipality is surprising and reason for major concerns regarding the design of the study. In the very least, this needs to be discussed.

13)   In the discussion, the authors state: “Our study was different, as we have an overall measure of whether the infant's motor skills are age-appropriate or not.” To me, this global measure is one of the main drawbacks of the study. All the detail and nuance of motor development is lost. It is a catch all measure and therefore catches nothing. I would appreciate a deeper discussion why being “different” in this regard is a benefit or what it tells the reader about motor development.

14)   Please add that you did not know the actual duration parents engaged with the materials and their child as a limitation.

15)   Similarly, the need for a longer term intervention should be moved to a “Limitations and future directions” section

16)   Could the different health advisors be included in your statistical model to control for assessment and scoring differences?

Round 2

Reviewer 2 Report

I would like to thank the authors for their responsiveness to my previous comments and for their fast turn-around on revising the manuscript. Overall, I do think the manuscript has improved and is clearer to understand now. I do have a few remaining suggestions for further improvement below. In addition, I would also encourage the authors to carefully proofread the entire manuscript for wording and language use. I did not see any incorrect use of English (except for maybe point 2 below), but some formulations are inefficient and could be improved. This is only a minor concern.

Concerns I would like the authors to address are:

1) On page two the authors now state that parents tend to overestimate their children’s motor skills. This is true, but other articles suggest that parents underestimate their true motor skills. And yet other researchers provide evidence that parents can accurately capture the child’s motor skills – especially when using diary measures. I recommend adding more of this nuance here. You may find this article relevant to cite or find some relevant citations to use in the article:

Libertus, K., & Landa, R. J. (2013). The Early Motor Questionnaire (EMQ): A parental report measure of early motor development. Infant Behavior and Development, 36(4), 833-842. https://doi.org/10.1016/j.infbeh.2013.09.007

In this article, the EMQ measure has been compared to direct observation measures and suggests that parents can be quite good at reporting the child’s motor skills.

The EMQ has also recently been adapted into the Polish language and a pre-print of this report can be found here:

https://assets.researchsquare.com/files/rs-2203517/v1/5a8a2c72-ad86-42a1-9d48-49b3b766a7e9.pdf?c=1668254376

I would encourage the authors to include some more of this nuanced discussion into the benefits of using a parent report measure vs. using an expert measure (as used here). Further, the authors could discuss how their expert measure is more limited in score compared to detailed assessment such as the EMQ but also the PDMS-2 or AIMS.

2) In Figure 2, you refer to “Comparising Municipality” – this should be “Comparison Municipality”. Please check the text throughout to correct this. Similarly, in Figure 2 you state “Hoeje-Taastrup” for the intervention group. This is inconsistent and the name of the municipality is actually irrelevant for the reader. Please change this to “Intervention Municipality” and “Comparison Municipality” throughout. The actual names of the municipalities should only be mentioned initially for context, but for clarity the intervention/control naming scheme should be used subsequently.

3) I disagree with the authors referring to this study as a “natural experiment” – as there was an intervention offered to one area but not the other. I assume the authors want to express that the two municipalities were not randomized into intervention vs. control conditions. I think the “natural experiment” term should be dropped, and the authors should state that the municipalities were a selected based on the planned implementation of the intervention and were therefore not randomized. This should also be mentioned as a caveat in the discussion section (you already do this when you state that the study is not an RCT).

4) In the conclusion, the authors speculate that the lack of improvement might be due to measurement error. As stated in my previous review, I do think the measurement is too “blunt” to identify more subtle changes in motor trajectories. This is not the same as having measurement error. The authors should rephrase this argument through the manuscript. The measure used here was rather global and holistic. It is possible that the intervention may have improved specific motor skills but that this was not detectable using a global rating.
